

# Citation.js: a format-independent, modular bibliography tool for the browser and command line

Lars G. Willighagen

Eindhoven, The Netherlands

## ABSTRACT

**Background**. Given the vast number of standards and formats for bibliographical data, any program working with bibliographies and citations has to be able to interpret such data. This paper describes the development of Citation.js (https://citation.js.org/), a tool to parse and format according to those standards. The program follows modern guidelines for software in general and JavaScript in specific, such as version control, source code analysis, integration testing and semantic versioning.

**Results**. The result is an extensible tool that has already seen adaption in a variety of sources and use cases: as part of a server-side page generator of a publishing platform, as part of a local extensible document generator, and as part of an in-browser converter of extracted references. Use cases range from transforming a list of DOIs or Wikidata identifiers into a BibTeX file on the command line, to displaying RIS references on a webpage with added Altmetric badges to generating "How to cite this" sections on a blog. The accuracy of conversions is currently 27% for properties and 60% for types on average and a typical initialization takes 120 ms in browsers and 1 s with Node.js on the command line.

**Conclusions**. Citation.js is a library supporting various formats of bibliographic information in a broad selection of use cases and environments. Given the support for plugins, more formats can be added with relative ease.

Corresponding author
Lars G. Willighagen,
lars.willighagen@gmail.com

## INTRODUCTION

All research extends or uses knowledge from other research. With the primary goal of scholarly publishing being the distribution of knowledge, it is important that the publications—and the literature they cite—are distributed in an accessible, identifiable and findable manner (*Shotton, 2013*). That also allows the analysis and visualisation of how research cites each other (*Shotton, 2013*; *van Eck & Waltman, 2014*). While traditionally journals required text-based citations, each formatted in their own specific style, the last few decades the use of Persistent IDentifiers (PIDs) has become commonplace, with Digital Object Identifiers (DOIs) being the most common for scholarly articles, and International Standard Book Numbers (ISBNs) for books. These PIDs are then linked to central stores that provide machine-readable bibliographic information, such as Crossref and DataCite (*Lammey, 2015*; *Brase, 2009*; *Neumann & Brase, 2014*).

Since most kinds of PIDs are intended for certain kinds of publication, be it data sets, journal articles, books or code repositories, the bibliographic information is stored in a format intended for that kind of publication. As a result, there are many different stores and many different formats: Libraries use Machine-Readable Cataloging (MARC) (*Avram, 2003*) and similar formats, Wikidata (*Malyshev et al., 2018*) and WikiCite (*Taraborelli et al., 2017*) have their own scheme; DataCite has DataCite eXtensible Markup Language (XML) and JavaScript Object Notation (JSON) (*De Smaele et al., 2017*); and Crossref has Crossref UNIXREF (*Crossref, 2018*). Similarly, most reference managers have their own formats too: Zotero and EndNote have their own schemes (*Vinckevicius, 2017*; *EndNote, 2012*), and Office Word has an XML namespace (*Microsoft, 2018*). On top of that there are a lot of old and new formats created for a variety of reasons, like BibTeX (*Patashnik, 1988*), Citation Style Language (CSL) (*Zelle, 2012*), Research Information Systems (RIS) (*Reference Manager, 2012*), and the Bibliographic Ontology (BIBO) (*D'Arcus & Giasson, 2009*).

This leads to reduced findability between organisations and disciplines (*Godby, Young & Childress, 2004*; *Zinn et al., 2016*), and reference managers need to maintain parsers for numerous formats and different types of citable resources (articles, books, data, software (*Smith, Katz & Niemeyer, 2016*), etc.). The management requires a detailed description of the source being referenced and preferably link to the full-text too (*Hull, Pettifer & Kell, 2008*). A second requirement is their ability to convert references into citations, according to the norms for formatting citations in writing (*Gilmour & Cobus-Kuo, 2011*). Reference managers assist in keeping references accessible and machine-readable, ready to be formatted for use in citation (*Fenner, Scheliga & Bartling, 2014*).

Existing managers, like Zotero, either require a client desktop program or a server, or have entirely proprietary backends. This paper introduces Citation.js, a standalone JavaScript library capable of running in the browser, on a server and as a CLI. It consists of a set of parsers and formatters (see Fig. 1) that together allow for the conversion of different metadata formats via a central format, CSL-JSON (*Bennett et al., 2018*). To better suit individual needs, and to minimize unnecessary code which is especially important in the browser, Citation.js is fully modularised. Formats are bundled in thematic plugins, which can be installed separately. For formatted bibliographies and citations, CSL styles and locales are used with `citeproc-js` (*Zelle, 2012*; *Bennett et al., 2018*). This paper describes how Citation.js is developed, documented, tested, and released.

## BACKGROUND

### Crosswalks

To convert one data format (or scheme) to another, a *crosswalk* is used. A crosswalk is a set of mappings between equivalent properties and entry types in different formats (*Pierre & LaPlant, 1998*). First of all, for most properties a simple mapping suffices: `title` in BibTeX refers to the same concept as it does in BibJSON and CSL-JSON. On top of that, the property coincides with `TI` in RIS. This mapping could come in the form of a JSON Linked Data (JSON-LD) context, as done by CodeMeta (*Jones et al., 2017*), as eXtensible Stylesheet Language Transformations (XSLT), as discussed by *Godby, Smith & Childress (2003)*.

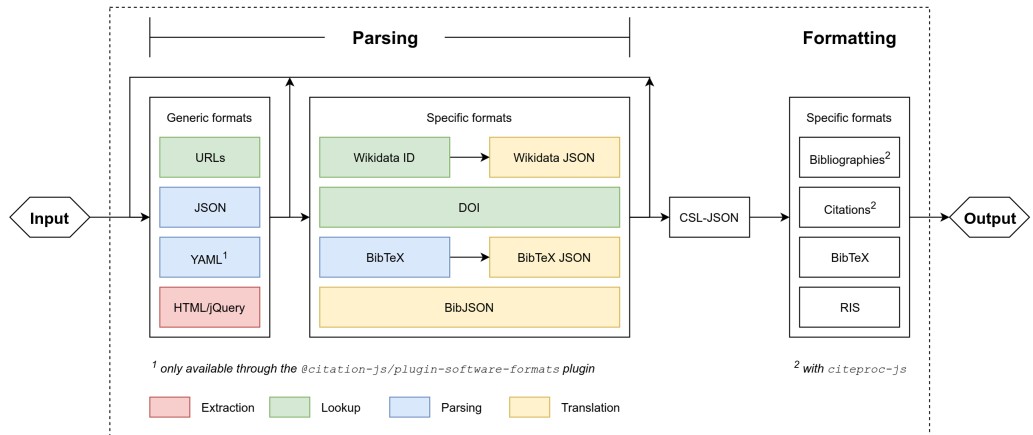

**Figure 1  Program setup of Citation.js.** Everything within the dotted square is part of Citation.js or its dependencies. Extraction is separating data from noise, look-up is fetching information based on a Uniform Resource Locator (URL) or PID, parsing is transforming text-based formats into data structures, and translation is transforming data structures with different schemas. Two types of output are supported: citations and bibliography, and machine-readable references.

Second, some mappings are context-dependent. For example, consider the CSL-JSON properties `author` and `reviewed-author` in relation to the RIS properties author (`AU`) and reviewer (usually `C4`). Normally, `AU` maps to `author`. However, if the entry being converted has the review entry type, `AU` maps to `reviewed-author` while `author` maps to `C4`.

Third, the data format of the values needs to be converted. While `title` in BibTeX can have formatting in the form of TeX, `title` in CSL-JSON uses a subset of HTML for formatting, and `TI` in RIS does not have formatting at all. Properties can also have different data types. In CSL-JSON, `author` is a list of objects, while `authors` in BibTeX, which describes the same concept, is serialized text delimited by ″ and ″.

Finally, there might not be a one-to-one mapping between properties. For instance, `page` in CSL-JSON maps to both `start` and `end` in Citation File Format (CFF) (*Druskat et al., 2018*). Similarly, `page`, `issue`, `volume` and `ISSN` are all top-level properties in CSL-JSON, while the corresponding properties in Wikidata are proposed to be nested in the journal property.

Since the last two aspects can lead to information loss, crosswalks often need to be one-directional converters between two formats. To not have to create crosswalks between every possible combination of supported formats, one could define a central format, similar to the "interoperable core" in the "long translation path" proposed by *Godby, Smith & Childress (2003)*. It is however important that the central format can hold as much information as should be represented in any of the output formats, as to prevent information loss when converting between two formats.

## Existing tools

Bibutils is very similar to Citation.js in that it also is a set of converters with a central format, there the Metadata Object Description Schema by the Library of Congress (*Putnam, 2005*).

Bibutils is used as a set of CLI programs, and does not directly allow formatting as citations and bibliographies. More recently, astrocite was created, a set of parsers that all output CSL-JSON (*Sifford, 2019*). Astrocite uses Abstract Syntax Trees (ASTs) and formal grammars, which should make it easier to write, read and maintain parsers. For drawbacks of formal grammars, see the Outlook.

The impact of reference managers should also not be underestimated. However, most reference managers have proprietary backends or require human input to use. Even Zotero, which is open source, is only commonly used via the client or alternatively as a server. While the fact that it has a server already allows for many more possibilities than other managers, it remains difficult to run a standalone program, which Citation.js does allow.

## APPROACH

As mentioned, Citation.js converts different bibliographical formats into each other. To achieve interoperability without too much work, a central format is chosen. All input is converted into this central format, and all output is created from it, adopting the approach of the "long translation path" with the "interoperable core" by *Godby, Smith & Childress, 2003*. See also Fig. 1.

Input is parsed iteratively: for each distinct format along the way from the input to the central format, a separate parser function is defined. This allow progressive enhancement, easily replacing parts of the parsing process without touching the rest, and lets users input intermediate formats without second thought.

Parsing iteratively is relatively simple, because all different kinds of input should get turned into a single type anyway: you do not have to choose what format you need next, you only have to recognize and parse what you have now. However, a similar process for output does not make as much sense. With output formatting, there is only one kind of input, the central format, and several kinds of output. If output formatting should be done iteratively as well, the paths to reach final output formats would have to be defined separately.

The Approach section is organized as follows. First, methods used while developing the software are listed. Then, design choices of the code itself, and how to install and use it is explained. Last, the method to evaluate the results is described.

### Software Development

The software was developed using modern standards: version control with Git, semantic versioning for releases (*Preston-Werner, 2013*), open source archives on GitHub (https://github.com/larsgw/citation.js; https://github.com/citation-js) and Zenodo (https://doi.org/10.5281/zenodo.1005176), browser bundles with browserify (*Halliday et al., 2018*), compatible code with Babel (*Zhu et al., 2018*), integration testing using the Travis-CI service (*Travis, 2018*), code linting (source code analysis) with ESLint (*Zakas et al., 2018*) and Standard (*Aboukhadijeh et al., 2018*), checking RegExp's for ReDOS vulnerabilities with `vuln-regex-detector` (*Davis et al., 2018*), and detailed documentation with JSDoc (*Williams et al., 2018*).

The development process took place with Node.js and npm. First off, any changes would be linted and tested with the aforementioned tools. Bugs or new features can also warrant the introduction of new test cases. If the changes work properly, they are then committed into the version control. If the changes warrant a new release, or if enough changes have piled up for a new release, the change log is updated. Updating the version in the package metadata automatically triggers the linters and test runners, preventing accidental mistakes. Afterwards, publishing the package to npm automatically triggers the generation of files necessary for the package. The scripts used for this are described in https://github.com/larsgw/citation.js/blob/90cd68c/CONTRIBUTING.md#installing.

## Libraries

Apart from tools used for development, Citation.js also uses a number of runtime libraries. Their function and the reason for using them is explained below.

| | |
|---|---|
| `@babel/polyfill` | is a runtime library which fills in support for modern APIs on older platforms. It comes with the use of Babel to transform modern syntax for older platforms (*Zhu et al., 2018*). |
| `citeproc-js` | is a widely used CSL formatting library written in JavaScript (*Bennett et al., 2018*; *Citation Style Language, 2018*). |
| `commander` | is a utility library, only used for the Command Line Interface (CLI). It parses the command line arguments and generates documentation (*Holowaychuk et al., 2018*). |
| `isomorphic-fetch` | is a specific polyfill, a library filling in support, for the Fetch Application Programming Interface (Fetch API), a modern way of requesting web resources. It works in both Node.js and browsers (*Andrews et al., 2018*). |
| `sync-request` | is a way to request web resources synchronously (*Lindesay et al., 2018*). While performing such operations synchronously is advised against in JavaScript, it is still useful for non-production scientific scripts, and demos. |
| `wikidata-sdk` | is a utility library for working with the Wikidata API (*Lathuiliére et al., 2018*; *Vrandečić & Krötzsch, 2014*) |

## Implementation

Citation.js employs a number of ways to achieve a balance between function and ease of use. The program consists of three major parts: the bibliography interface, code handling input parsing, and code handling output formatting. The bibliography interface itself is quite simple; it mainly acts as a wrapper around the parsing and formatting parts. These two parts behave in a modularised way, with a common plugin system.

### Input parsing

Input parsing works by registered input formats. These registrations include an optional type recognizer and a synchronous and/or an asynchronous function transforming the input into a format closer to the final format: CSL-JSON. The new input can then be tested again, and will be parsed iteratively until the final format is reached. Plugin authors are

encouraged to create input parsers with as small steps as possible, to allow users to input a variety of different formats.

Type recognition is done with a search tree. First of all, types are ordered by the data type of the input. This is one of: String (unparsed text, identifiers, etc.), SimpleObject (standard JavaScript Object), Array (a possibly non-uniform list of inputs), ComplexObject (other non-literal values) and Primitive (numbers, null, undefined). The data type can be inferred from other format specifications in some cases. Types can also be specified to be a more specific version of something else. For example, a DOI URL is also a normal URL, but should be parsed differently, namely with content negotiation.

Types can then provide a list of predicates, testing if input belongs to that format. To avoid code repetition and make plugin registration code easier to read, certain common tasks can also be accomplished using shortcuts. These shortcuts include testing text against a RegExp pattern, checking for certain properties and checking for the value of elements in an array. These properties can also eliminate the need for an explicit data type: for example, if a RegExp is provided, input can be expected to be a String.

### Output formatting

Output formatting is less complicated. Users and developers only have to provide the identifier of the formatter. Further customization can then be done by providing options, which are automatically forwarded to the formatter. This allows the CSL plugin to take in options specifying the template and locale, for example. All formatting producing bibliographies and citations is done with citeproc-js (*Bennett et al., 2018*).

### Plugin system

Apart from being able to add input and output formats and schemes on their own, it is also possible to add them in a thematically linked plugin. For example, a BibTeX plugin might consist of a parser for .bib files, a parser for the resulting BibTeX-schemed JSON, and an output formatter to create BibTeX from other sources as well. This plugin could then be combined with, for example, a Bib.TXT plugin, resulting in a JavaScript package or module, which could be published in package managers like npm. Code for this plugin would look like Fig. 2.

For configuring plugins there is also a config option. As an example a labelForm option is added, which could control the way the BibTeX output formatter generates labels. Users of this plugin can then retrieve and modify this configuration. It is also possible to offer internal functions this way, for more fine-grained control.

### Bibliography interface

The methods for parsing input and formatting output are also included in a general class, Cite. Class instances also have access to opt-in version control—changes are tracked if an explicit flag is passed—and sorting. The latter currently does not have effect on CSL bibliographies unless set with the nosort option, as the styles define their own sorting method.

```
1   let Cite = require('citation-js')
2
3   Cite.plugins.add('bibtex', {
4     input: {
5       '@bibtex/text': {
6         parseType: { ... },
7         parse (text) { ... }
8       },
9       '@bibtex/object': {
10        parseType: { ... },
11        parse (text) { ... }
12      }
13    },
14
15    output: {
16      bibtex (data, options) {
17        ...
18      }
19    },
20
21    config: {
22      labelForm: ['author', 'title', 'issued']
23    }
24  })
25
26  let bibtexConfig = Cite.plugins.config.get('bibtex')
27  bibtexConfig.labelForm = ['author', 'issued', 'year-suffix']
```

**Figure 2** **Possible structure of a plugin for BibTeX.** In this example package, line 1 loads Citation.js and lines 2–24 adds the plugin. This plugin consists of two input formats (4–13), one output format (15–19) and configuration options (21–23). Lines 26–27 show how this configuration would be used. Some code is omitted for the sake of clarity, and is replaced with ellipsis (...).

**Table 1** **Input and output format support.** This table only shows general support. For example, the "Wikidata" format is both used for Wikidata identifiers and Wikidata API results.

| Format | BibJSON | BibTeX | Bib.TXT | CSL | DOI | RIS | Wikidata |
|--------|---------|--------|---------|-----|-----|-----|----------|
| Input | x | x | x | (JSON) | x | | x |
| Output | | x | x | x | | x | |

### Supported formats

Table 1 shows the formats supported by Citation.js at the moment.

## Distribution
### Browser use

For in-browser use, there is also a standalone JavaScript file available. This includes dependencies. This bundle is built automatically when publishing, and is available through a number of Content Delivery Networks (CDNs) that automatically distribute npm packages. The Cite class can then be imported and used just as the npm package, barring browser limitations.

For simple use cases like inserting static bibliographies, a separate tool, citation.js-replacer, was developed. When included on a page, this replaces every HyperText Markup Language (HTML) element matching a certain selector with a bibliography.

```
1   <html>
2     <head>
3       <!-- Altmetric widget code --> <script
          ↪ src="https://d1bxh8uas1mnw7.cloudfront.net/assets/embed.js"></script>
4       <script src="https://cdn.jsdelivr.net/npm/citation-js"></script>
5     </head>
6     <body>
7       <div id="element"></div>
8       <script>
9         window.onload = async function () {
10          let Cite = require('citation-js')
11          let cite = await Cite.async('10.1371/journal.pone.0185809')
12
13          let bibliography = cite.format('bibliography', {
14            format: 'html',
15            append ({DOI}) {
16              return ` <span class="altmetric-embed" data-badge-type="badge" data-doi="${DOI}"></span>`
17            }
18          })
19
20          let element = document.getElementById('element')
21          element.innerHTML = bibliography
22          _altmetric_embed_init()
23        }
24      </script>
25    </body>
26  </html>
```

**Figure 3** **Basic use, including appending data to formatted bibliography entries.** Here, line 3 loads the Altmetric widget code, line 4 loads the library, line 10 imports `Cite`, and line 11 creates an interface for a bibliography with one entry, with metadata from a DOI. Lines 13–18 render the bibliography, with line 14 setting the output to HTML and lines 15–17 appending an Altmetric widget to the entry. Lines 20–21 show the output on the page. Lines 9 and 23 are to avoid race conditions in DOM access. Lines 3 and 22 initialize the Altmetric badge. In the example, (*Hallmann et al., 2017*) is used.

Hallmann, C. A., Sorg, M., Jongejans, E., Siepel, H., Hofland, N., Schwan, H., … de Kroon, H. (2017). More than 75 percent decline over 27 years in total flying insect biomass in protected areas. *PLOS ONE, 12*(10), e0185809. https://doi.org/10.1371/journal.pone.0185809 [Altmetric 5,793]

**Figure 4** **Result of the code in Fig. 3.** Bibliography consisting of *Hallmann et al. (2017)* in APA style. Note the Altmetric badge at the end.

Figure 3 shows an example of another use case. For example, the basic use can be extended to add additional information to citations, such as an Altmetric (*Adie & Roe, 2013*) score icon or Dimensions citation count (*Thelwall, 2018*). The output is shown in Fig. 4.

### npm package

Citation.js is published as an npm package on the main npm registry, as `citation-js`. Use of the package is the same anywhere, apart from platform limitations. For example, synchronous requests for web resources, used to get metadata for DOIs, is limited on Chrome as discussed in *Willighagen (2017b)*. Also, the Node.js platform, not being a browser, doesn't have access to the Document Object Model (DOM), and so can't easily use HTML elements as input or output.

Separate components, including formats not included in the standard configuration are available under the `@citation-js` scope.

Use cases for the npm package include using it when generating content (either at runtime or for static websites) like PubPub (*Shihipar & Rich, 2018*), and setting up APIs (*Willighagen, 2017a*). It is also useful for converting metadata when text mining. For example, BibJSON is one of the input formats, and can then be converted to BibTeX or

formatted. All references for GitHub projects were created with a simple script running Citation.js.

### CLI use

Simple one-time conversions, with no extensive customization, can also be done with Command Line Interface (CLI). The command can be installed with npm, which may require root privileges depending your setup. Alternatively, any commands can be prefixed with `npx` instead. The command can get input text from files, command line arguments or via standard in. Output can be configured with a number of options detailed in the man file, also available by running with the `-h, -help` option. Any output is then written to a file or redirected to standard out.

### Integrations

The Citation.js npm package can also be used as a library to create integrations with, among other things, word processing systems. For example, ReLaXed (*Zulko et al., 2018*) integrates Citation.js into the Pug templating language to generate citations when creating Portable Document Format (PDF) documents, and the npm package `citation-js-showdown` was created as a demo on how to introduce syntax for citations in Markdown.

## EVALUATION

### Experimental setting

#### Coverage

Coverage of types and properties was determined by creating two spreadsheets, one with all of the CSL types and one with all of the CSL variables. Then, columns where created for other supported formats and filled in with the corresponding type or property in that format. The amount of mappings were counted as a percentage of the total possible mappings, i.e., the total amount of types or properties, available in CSL.

This method of counting skews the perspective as not all properties and types can plausibly be mapped, either because no equivalent term exists in the other format, or because the existence is currently unknown to the authors. This is explained in more detail in the results.

#### Impact

To collect dependent projects, the GitHub Dependency Networks was used, which lists other GitHub repositories listing Citation.js as a dependency. To find dependent repositories on different hosting platforms, a search on the respective hosting platforms and Google was carried out. Additionally, projects known to the authors to use the library were listed. Of those lists, a diverse set of projects was extracted by hand, followed by a check to see if and how Citation.js is used.

For download counts, `npm-stat.com` was chosen because of its ability to collect download statistics over specific, multi-year time frames.

#### Performance

Performance statistics were gathered by recording runtime performance in the Chrome DevTools and the Firefox Developer Tools while importing Citation.js in the browser. The

**Table 2  Mapping statistics.** The number of CSL-JSON properties and types mapped to different formats. See Table S1.

|  | CSL-JSON | Wikidata | Wikidata (>0.4.4) | BibTeX | RIS | BibJSON |
|---|---|---|---|---|---|---|
| **Properties** | 78 | 24 (31%) | 46 (59%) | 21 (27%) | 42 (54%) | 20 (27%) |
| **Types** | 35 | 31 (89%) | 31 (89%) | 11 (31%) | 30 (86%) | 11 (31%) |

results were obtained with the default settings for each of the platforms. In Chrome, this was achieved by using *guest mode*, while no custom settings were used for Firefox and Node was used without command-line options affecting performance.

The sizes of the different components was determined with the disc tool (*Kennedy et al., 2019*).

## Results
### Coverage
An important aspect of Citation.js, other than parsing and formatting specific syntaxes, is the mappings between different formats. Since creating mappings between each format is often unnecessarily much work, CSL-JSON was chosen as a central format. To review the existing mappings, the number of mapped properties and entry types relative to CSL-JSON were counted. Table 2 shows those results.

While the numbers may seem low, note that not every CSL-JSON property can be mapped: the intentions behind at least three properties are contested (*Wiernik, 2018*), the values of two other properties can usually be derived from other fields, ten properties are specific to references and may not apply to resource-describing schemas, and twenty properties could be reduced to just eight with linked data. Additionally, a number of properties have limited documentation and usage, making it difficult to determine what the exact meaning is.

On top of that, the other formats may not have enough well-defined properties to map either. The BibTeX and BibJSON mappings, the latter being based on BibTeX, are limited by the low number of known properties and types. Without an authoritative list of properties, examples from a range of sources were used to define a mapping, which consequently lacked lesser-used properties. The Wikidata mapping has the most potential for expansion due to the large number of described properties. In fact, a recent Citation.js update nearly doubled the number of mapped properties to 46 (59%). At that point, the CSL specification becomes limiting again.

The number of mapped RIS properties is actually higher than expected; the RIS mapping has a lot of one-to-many and many-to-one mappings, which makes it inconsistent while artificially raising the number of mapped properties. Since there is no authoritative document other than an Internet-Archived spreadsheet linked to on Wikipedia (*Reference Manager, 2012*), the current mapping is partially based on the Zotero translator for RIS.

Apart from property and type mappings, value conversion affect the accuracy of results as well. For example, since BibTeX does not encode information about how names are built up, such information has to be estimated. While this is seems to be going fine for Western names, other naming systems may not work as well. The same goes for how RIS encodes

**Table 3   Different uses of Citation.js found in the wild.** Some of these projects contributed valuable feedback to the development of Citation.js.

| User | Use | Parsing | Formatting |
| --- | --- | --- | --- |
| wcite (*Voß, 2019*) | CLI tool for managing a Wikidata bibliography | Wikidata | Yes |
| Reference Extractor (*Zelle & Zumstein, 2018*) | Browser tool extracting references from Word documents | No | Yes |
| PubPub (*Shihipar & Rich, 2018*) | Live server-side generation of citations (with REST API) | No | CSL, BibTeX |
| service-based-antipatterns (*Boceck, Popp & JREB, 2019*) | Live in-browser generation of citations | No | CSL, BibTeX |
| ds.korea.ac.kr, ccv.brown.edu | Static site generation of lists of publications | BibTeX/DOI | CSL |
| dabai.compute.dtu.dk | Live in-browser generation of lists of publications | Wikidata | CSL |
| schol-js, RelaxedJS (*Czerwinski, 2018*; *Zulko et al., 2018*) | Document generation | Yes | Yes |
| Ovide, Fonio (*de Mourat & Rabot, 2019*; *de Mourat, Plique & Pichon, 2019*) | Experimental publishing platform | BibTeX | CSL |
| PolarisOS (*Ribeyre, Louis-Marie & MyScienceWork, 2019*) | Library management system and repository | No | CSL |

names: a first name, a last name and an optional suffix. Note, however, that Zotero, which uses CSL, is not particularly geared towards any naming schemes other than the most simple, as noted by *D'Arcus (2008)*. Also, style guides themselves may not have rules for non-Western names either (*Qiu, 2008*; *Puniamoorthy, Jeevanandam & Narayanan Kutty, 2008*).

### Impact

Since Citation.js was published as an npm module, it has been used independently of the authors in a variety of use cases. With the GitHub Dependency graph, Google, and via Twitter, a number of those uses can be identified, listed in Table 3. The download count is also increasing, as can be seen in Fig. 5. Between the first version in October 2016 and 26 April 2019, our package was downloaded a total of 26,718 times.

### Performance

The performance of the Citation.js package has been analyzed on a number of different platforms. Between browsers, compiling the script and importing the library takes about 120 ms, compiling itself taking a little less than half of that. Node.js on the other hand takes about 1 s to initialize, both when the source consisting of multiple files is imported, and when a bundle is imported. This is possibly because Chrome caches compiled JavaScript reducing the compiling times from around 50 ms to about 8 ms, as is explained in *Alle (2018)*.

As shown in Fig. 6 and Table 4, time taken to import the library mainly consists of importing @babel/polyfill. This is because adding the polyfills requires repeated feature detection. After that the actual code is imported in two parts. In the first part, where core functionality like the Cite interface is loaded, the main culprit is addTypeParser, with 0.13 ms per call on average. In the second part, loading output-related code, importing citeproc-js takes the longest with a single call of 2.82 ms. Note that that Firefox uses Just-In-Time (JIT) compilation, compiling pieces of code when they are used a lot.

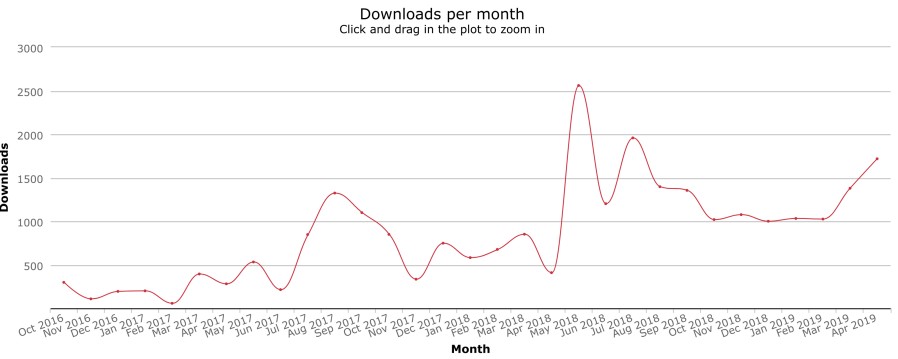

**Figure 5** **Download counts since package creation.** Graph from npm-stat.com, data from npm. See Data S1.

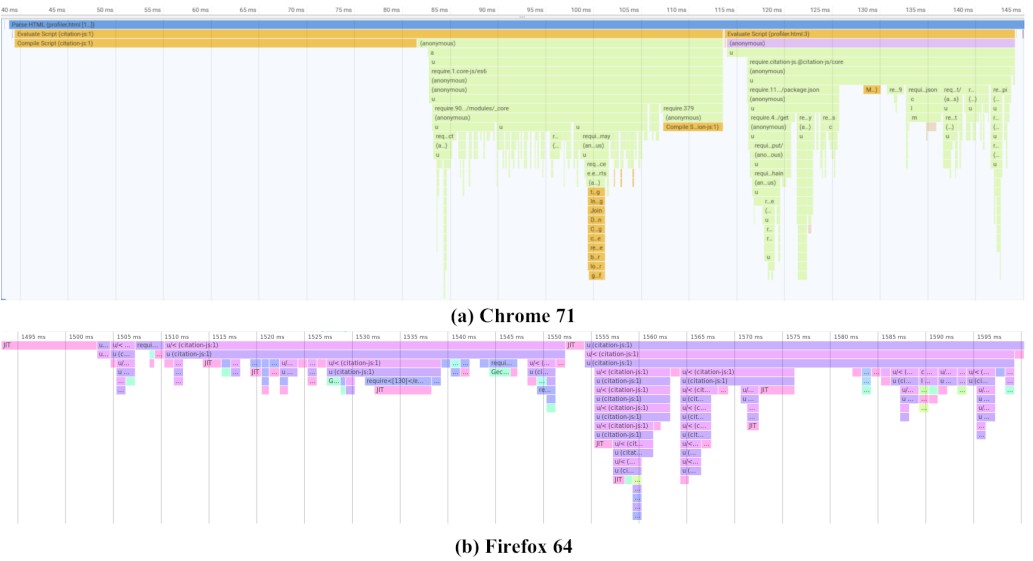

(a) Chrome 71

(b) Firefox 64

**Figure 6** **Initialization performance results on different platforms.** Actual timings may vary depending on the device, operating system and cache. Note that Chrome (A) starts with 40 ms of compiling time that is cached on subsequent runs. Firefox (B) compiles JIT, while the code is running. Both graphs show three parts, one loading polyfills from `@babel/polyfill` taking up half the loading time, followed by two parts mainly loading core functionality and plugins respectively. Profiling data is available as Data S2 and Data S3.

While code execution is one part, one should also look into the file size. This is especially important in the browser, which has to fetch the library when loading the page. The biggest part is `citeproc-js`, accounting for almost half of the file size. Additionally, built-in CSL styles and locales should also be counted. A complete overview can be found in Table 4.

**Table 4  Browser bundle breakdown.** Running time is the time it takes to import that part of the script with browserify `require` in the Chrome data set. Note that a big part of the `plugin-csl` "own" code is serialized styles and locales from the CSL repository. Minified the code is 702 kB, which is reduced to 177 kB with gzip and 164 kB with Brotli, both with default compression levels.

| | | Size | | | |
|---|---|---|---|---|---|
| | **Part** | **Own** | **Dependencies** | **Total** | **Running time** |
| | Backport | 5.9 kB | – | 5.9 kB | 5.9 ms |
| | core | 99.7 kB | 33.9 kB | 133.5 kB | 8.3 ms |
| | plugin-bibjson | 7.6 kB | – | 7.6 kB | 3.5 ms |
| | plugin-bibtex | 42.9 kB | – | 42.9 kB | 2.6 ms |
| Citation.js | plugin-csl | 87.0 kB | 461.8 kB | 548.9 kB | 3.2 ms |
| | plugin-doi | 6.6 kB | – | 6.6 kB | 0.6 ms |
| | plugin-ris | 11.1 kB | – | 11.1 kB | 0.5 ms |
| | plugin-wikidata | 22.1 kB | 40.4 kB | 62.4 kB | 2.2 ms |
| | name | 16.6 kB | – | 16.6 kB | 1.1 ms |
| | date | 7.3 kB | – | 7.3 kB | 0.2 ms |
| Additional | @babel/polyfill | – | 197.3 kB | 197.3 kB | 32.0 ms |
| | browserify | – | 6.8 kB | 6.8 kB | 0.2 ms |
| Total | | 306.8 kB | 740.1 kB | 1046.8 kB | 60.3 ms |

## DISCUSSION

### Converting between formats and standardized crosswalks with linked data

Converting input data like parsed BibTeX, BibJSON or Wikidata API results into another format and back can get very repetitive in terms of code. Yet, there are still cases where special handling is needed. Since different formats call for different needs, each plugin has developed its own system to deal with this. Unifying this into a single, performant, reusable and developer-friendly system would be preferable.

*Jones et al. (2017)* use a JSON-LD context for this. While a JSON-LD context would scale very well without a central format, most cases restrict the usefulness. Consider the `page` property in CSL-JSON, mapping to the `first` and `last` properties in CFF. If the nested values were deserialized, this could be expressed in JSON-LD contexts. However, in the first example it cannot, since JSON-LD cannot distinguish between parts of strings.

Alternatively, a custom system could be developed that defines as much mapping as possible to and from a central format, with special cases for context-dependent mappings and one-to-many mappings. It would be difficult to do this entirely language-agnostic, since serialization and deserialization usually requires some amount of scripting.

### CSL-JSON as a central format

As mentioned in the Background, an important feature of the central format is that it can hold any information needed for the output formats—if it cannot, information loss can occur. Citation.js currently uses CSL-JSON as a central format, as it has a (mostly) well-defined list of properties and entry types, an authority to clear up any confusion, and relatively good support for most metadata while still being simple to work with.

However, CSL-JSON is not perfect, and information loss is definitely possible. This is currently the case with the software entry type, which does not exist in CSL 1.0.1, and is represented by the book type by convention. When converting Wikidata input consisting of a computer program to RIS output, the fact that it is in fact a computer program is lost, even though both formats support it as an entry type.

A solution for this would be to extend the format to allow for the missing entry type or property, assuming the specification authors agree. Otherwise, a custom extension could be made. Doing that for every future shortcoming may not be sustainable though, since it effectively creates a new poorly-supported standard to add to the mix. Choosing a different format is among the possibilities, but we have not found a suitable candidate; even Wikidata, which is intended to cover everything, is changing constantly and even if a proper specification is created, the question remains whether the bulk of the publication data is up to date.

If no suitable format is found, one might add additional mappings to common formats, to cover properties missing from the central format. Then, find a path for every property through the crosswalks to get to the end format. That way, one could bypass the central format in specific cases only, keeping an easy mapping for common properties.

### Scraping from source versus fetching from central stores

When getting data from, for example, the Wikidata API or scraped from a web page, that data may be incomplete. However, if part of the data you get is the DOI linked to the entity queried, you could amend that data with data fetched from a central store like Crossref or DataCite. Due to difficulties with prioritizing data sources and non-trivial merging conflicts this has not been implemented yet, although linked-data formats such as Resource Description Framework (RDF) and JSON-LD would be possibilities. Additionally, if the user specifically requests data from a specific API, it can be assumed they want that specific data to be used.

## OUTLOOK

### Use of formal grammars for parsing

Apart from more common formats like JSON, XML and YAML (YAML Ain't Markup Language), Citation.js has to parse a number of text formats with syntax specific to that format, like BibTeX and RIS. While one can use standard or even built-in parsers for common formats, that is usually not possible for the latter formats. To solve this, one can employ formal grammars, which can be translated into code parsing and validating input. Examples of libraries working with grammars are PEG.js (*Majda et al., 2018*) and nearley.js (*Kartik et al., 2018*). Creating grammars has the benefits of not having to write and maintain code validating and parsing input, and having a readable grammar instead of a complex program file.

However, there are also drawbacks. Generating these grammars requires an extra build step, which in the case of Citation.js cannot be integrated with the preceding step due to a lack of appropriate tooling. On top of that, early tests have shown generated code to have poor performance or large size, and in the case of nearley.js requires a runtime library.

Therefore, it may be preferable to write custom parsers in some cases. For example for RIS, which has no balanced brackets or quotes, and does not need much more than simply iterating over the individual lines.

### Use of GraphQL for API queries

With REpresentational State Transfer (REST) APIs comes the problem of over-fetching and under-fetching: when fetching a resource it may contain too much unneeded information, require additional calls to the API, or both. This causes unnecessary load on both the client and the server, as both have to process more calls with more network bandwidth. One case where this is especially relevant is Wikidata, which in its linked-data nature does not expose the name of the authors or journals it links to; to retrieve that, additional requests are needed. To overcome this a better way of generating those requests locally could be implemented, or a GraphQL server could be developed by allowing the client to specify exactly what data it needs (*Facebook, 2018*).

### Support for additional formats

Apart from the formats currently supported in Citation.js (see Table 1), there are plans to include more formats such as EndNote import files, MARC XML (*Avram, 2003*), the Zotero API JSON schema and Office XML. These will be published in thematic plugins. For example, formats used to describe software projects are joined in the plugin `@citation-js/plugin-software-formats`. These formats will also include linked data scraped from web pages.

## CONCLUSIONS

Citation.js has been introduced as a library that supports bibliographic information in various formats, from multiple sources. The use of JavaScript ensures it can be used in a wide variety of use cases in the web browser, on the command line, and in a server environment. The tool is developed using modern approaches and released via the npm network and archived on GitHub and Zenodo. In addition to machine-readable formats such as BibTeX and RIS, the support for CSL styles ensures that citations and bibliographies can be formatted in many textual representations. Additional content can be easily added to those representations, such as Altmetric icons. The support for plugins allows additional formats to be integrated with relative ease, and without the need of a central repository managing those plugins.

## ACKNOWLEDGEMENTS

Thanks to JS.org for providing the (sub)domain name for the homepage of Citation.js. Thanks to the many people submitting bug reports, pull requests, and other kinds of feedback during the development of Citation.js.

### Funding

The author received no funding for this work.

### Competing Interests

The author declares there are no competing interests.

### Author Contributions

- Lars G. Willighagen conceived and designed the experiments, performed the experiments, analyzed the data, contributed reagents/materials/analysis tools, prepared figures and/or tables, performed the computation work, authored or reviewed drafts of the paper, approved the final draft.

### Data Availability

Source code is available on GitHub at the https://github.com/larsgw/citation.js repository and the https://github.com/citation-js organisation, and on Zenodo:

Lars Willighagen, Egon Willighagen, The Gitter Badger, Petr Čermák, & Johannes Wienke. (2018, November 2). larsgw/citation.js: v0.4.0-10 (Version v0.4.0-10). Zenodo. http://doi.org/10.5281/zenodo.1476934.

The library is distributed on npm at https://www.npmjs.com/package/citation-js and the individual modules in the https://www.npmjs.com/org/citation-js organisation.

### Supplemental Information

Supplemental information for this article can be found online at http://dx.doi.org/10.7717/peerj-cs.214#supplemental-information.

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
