# Peer review of "Citation.js: a format-independent, modular bibliography tool for the browser and command line"

_PeerJ Computer Science, doi:10.7717/peerj-cs.214_

## Round 0.1 · original submission · Major Revisions

Dear author,

I have carefully evaluated the three reviews of the article. All the reviewers explicitly said that the tool introduced in the article is great. However, they also identified several flows in the narrative that must necessarily be taken into account – which means to drastically extend and revise several parts of the content of this paper. I believe this can be done within a (extensive) major revision round. I suggest to the author to keep all the reviews in careful consideration for revising the article. Here a summary of the major issues identified:

1. This is not a typical research article since it actually introduces a tool, which is totally fine and I think these kinds of contributions should be accepted in PeerJ CS as well – something I've already done in the past. However, this means that the usual proposed argumentative structure does not work well in this case. I really appreciated the comment by Reviewer 2 who pointed to the ISWC guidelines for resource papers (see https://iswc2019.semanticweb.org/call-for-resources-track-papers/), which can be very helpful for revising the text. See in particular the review criteria at that page, that actually are greatest guidelines also for authors. Thus, I urge to revise the article according to these guidelines so as to make the article more sound for the kind of content it describe.

2. An important part that should be addressed, since it is missing in the article, is a related works section where existing works working along the lines of the tool presented in this document are also described and, possibly, compared. This is something that has been indeed requested by all the reviewers. In addition, it would be good to provide also a more general setting of the work by expanding the background section to provide more details.

3. In the evalutation of the tool, in addition to the study on performances that has been already conducted, and which is interesting, it is needed to discuss the current known uses of the tool. This is a crucial aspect to address since it is the way one understand the community uptake about it. Also, mentions to possible places where the tool can be adopted and which advantages can bring should be discussed as well.

4. The "Results" and "Discussion" sections must be carefully revised, since they contain a lot of content that should not be present there but rather in other sections.

Of course, please take into consideration all the suggestions highlighted in all the reviews. Thus, as anticipated, I would suggest to have a "major revision" as final decision.

Thanks again for submitting your contribution to PeerJ CS.
Have a nice day :-)

S.

·

Basic reporting

The article gives a detailed overview of the Citation.js software with impressive results. The article is a more a report than a research paper and it provides a practical solution to the relevant problem of accessing, and converting bibliographic data for citation. The background section should slightly be extended with references to related works (see comments on validity of the findings). Apart from that there are only minor issues, listed here in their order of appearance :

- subsection "output formatting" (line 104) could make more clear that the most of the formatting is done via the existing library citeproc

- subsection "CLI use" (line 152) could be reduced as it contains general information common to most command line application

- figure 3 is too large

- there seems to be a "not" missing between line 215 and 216

- in the same subsection one might mention that extension of the crosswalk format CSL-JSON might be a solution

- the subsection on GraphQL (line 235-242) should be removed as it its currently not used in Citation.js and the specific relevance is not made clear

- in section line 249-252 one might mention RDF as it is better suitable for merging data from multiple sources that other formats

Experimental design

no comment

Validity of the findings

The results are relevant and created in the best form by modern software development, but the research question has been addressed before. The article in should be extended by references to existing solutions and related works. In particular:

- mention library catalogues as central stores of bibliographic data (line 29) with library-specific formats such as MARC

- mention existing tools: At least Chris Putnam's Bibutils is relevant here: Citation.js seems like a modern successor to Bibutils

- add a subsection about the problem of converting between bibliographic data formats in the background section (with reference to subsection starting at line 198 for details). The topic of schema/metadata crosswalks is relevant both in general information integration and in metadata management in particular. The article does not need to give a full review (see https://en.wikipedia.org/wiki/Schema_crosswalk for a starting point) but it should briefly describe where crosswalks between bibliographic formats already exist (for instance XSLT has played a major role).

·

Basic reporting

This paper introduces citation.js, which seem to be a very useful and important tool to map between the different formats currently used to represent citation information. The paper is relatively easy to read, but unfortunately not written and structured in a very scientific way and lacks crucial elements. The reader is given many details about the technical details of the tool, but unfortunately virtually nothing is said about its position with respect to the existing scientific literature on the topic of citations, and very little in terms of evaluation results.

Specifically, the main sections Background, Results, and Discussion are all very unconventional and lack important scientific content:

- The Background section only introduces technologies and libraries that were used to build the tool, but not a broader background in a scientific sense. I expected to read something about other work on analyzing citation networks, on the usage and impact of reference managers, on the properties and uptake of ontologies on scholarly communication, just to mention a few examples from the top of my head.

- The Results section contains mostly details about the implementation of the tool, which are not results in the scientific sense. These implementation details should rather go into a section called Approach or Implementation or Methods.

- The performance evaluation is the only real result in the Results section, but performance is in my view a secondary problem here. More importantly, I would like to see some results on whether the tool is accurate, useful, and/or actually making an impact.

- The Discussion section is also much too technical, whereas I would expect here some more high-level context and interpretation of the presented results.

In general, the paper seems to be more of the kind of a Resource Paper as some conferences and journals have them (e.g. here http://iswc2018.semanticweb.org/call-for-resources-track-papers/), but these papers are typically required to demonstrate and quantify adoption, or at least to provide "evidence and motivation for claiming potential adoption". PeerJ doesn't seem to have an explicit call for such resource papers, but this paper would in any case not meet such typical Resource Paper requirements either.

Therefore, in summary, I think that this paper shouldn't be accepted. I have the impression that citation.js is indeed a very useful and important tool, but the omissions of the paper as explained above are too serious to warrant Major Revision as a decision.

Experimental design

No experimental design is reported.

Validity of the findings

No scientific findings are shown, except for performance measures, which are of secondary nature.

·

Basic reporting

This is a well written presentation of an important and impressive open-source software contribution and as such I'd recommend it for publication in PeerJ. Most of my review is contained as comments in the uploaded PDF, but some general comments:

I don't think the article is well-served by following a structure designed for more "traditional" empirical articles. E.g., it wasn't clear to me why some things were "Results" and some were "Discussion". I would revise this structure, and my reading of PeerJ guidelines allow, if not encourage, authors to do so.

Experimental design

no comment (does not apply)

Validity of the findings

no comment (see PDF)

Additional comments

The biggest advantage of a journal publication for a software package over simply providing the DOI for the source code itself (which is already the case for citations.js) is that it allows authors to explain their choices and practices in developing the software and to provide a "bigger picture". The present article fully delivers on the former, but I'd like to see some more big picture. For one, a clearer exposition of what exactly the tool does, right at the start of the paper, would help a lot. Secondly, a list of current implementations would be helpful -- I know there's at a minimum Wikicite, but I figure there are others? As it currently stands, the articles requires the reader to be very familiar with the metadata/citation landscape to understand the tool's contribution. I think this can be explained in a way that most readers familiar with academic referencing more broadly could be convinced of its importance.

---

## Round 0.2 · Minor Revisions

Dear author,

Thanks for your revision. It already improves a lot the quality of the paper. I think the paper is now positioned better than its initial submission and, surely, it is in the right track for being accepted.

However the reviewers, in particular one of them, have raised some points that still need to be appropriately addressed in the content, so as to make the paper strong enough to finally deserve to be accepted in PeerJ CS. Thus, please take into consideration all his suggestions and comments and carefully address them.

Thanks again for your revised submission. I’m really looking forward to receiving your next revision, which I’m sure will be great and will address all the issues raised, making the article an incredible contribution to PeerJ CS.

Have a nice day :-)

S.

·

Basic reporting

With its focus on a scientific tool, the article is more a report than a research paper. It provides a practical solution to the relevant problem of accessing, and converting bibliographic data for citation. The article gives a detailed overview of the Citation.js software with impressive results. The software is well described with motivation, capabilities and compared with existing solutions.

Experimental design

The article does not describe a scientific experiment but the outcome of engineering. Motivation and goals are clearly described and the article includes some technical evaluation. Additional user studies may be appropriate if the software was more end-user oriented but it is not required in this case.

Validity of the findings

The results are relevant and created in the best form by modern software development. The software documentation is better than normal at current research software so the article can also help to increase awareness for other kind of scientific software reports. The documented mapping table between CSL JSON and other bibliographic formats is valuable as research data in its own right.

Additional comments

Thanks for the thorough extension of the article.

·

Basic reporting

Quite well written. Background needs to be improved (see below).

Experimental design

Evaluations make sense, but need to be explained better (see below).

Validity of the findings

The findings are quite technical but seem valid (see below).

Additional comments

This paper has greatly improved from the last revision. It is, however, still not written in a sufficiently scientific way, with too much focus on technical details and not enough high-level explanations and motivations.

Major comments:

- The Background section has now a different content than before, partly copied from other places of the previous manuscript, but it is still not a proper background section. It is very technical and very narrow, and thereby fails to give a general and broad overview of this research field. Specifically, this comment from my previous review still holds: "I expected to read something about other work on analyzing citation networks, on the usage and impact of reference managers, on the properties and uptake of ontologies on scholarly communication, just to mention a few examples from the top of my head."

- As with the Background, also the Approach section lacks a high-level description. The content is all very low-level and technical, which is OK for such an Approach section, but only if preceded by a high-level introduction and explanation. The Approach section start with pointing out what version control system was used, instead of introducing the high-level Citation.js approach.

- I like the addition of the accuracy reports in the Results section. This really strengthens the paper! However, "accuracy" is highly confusing as a term, both intuitively and technically (because of its very specific meaning in statistics). Reporting accuracy levels of 27% makes your tool look performing very poorly when the common interpretation of "accuracy" is applied. I suggest to rename "accuracy" with a more appropriate term, e.g. "coverage". Moreover, the text could make it a bit clearer to what extent these numbers come from a lack of expressiveness of the studied formats or a lack of implemented features on the side of Citation.js (I think it's already in the text to an extent but I found it difficult to understand).

- The designs of the evaluations presented in the Results section should moreover be introduced better (either in a previous section such as Approach, or before the actual results are described in Results). Now, these designs are only described implicitly together with the results. It is good style to separate the two when describing evaluations, i.e. first describing the evaluation's design and only then its results.

More minor comments:

- I like Figure 1! But it's a bit unclear which parts make up the Citation.js tool (all the arrows? only the arrow pointing from "Input" to "Output"?)

- I like the addition of Impact in the Results section! I think it would help further if Figure 5 was quickly summarized in a sentence (like "Overall, our package was downloaded X times since its creation in October 2016").

---

## Author Rebuttal · Round 0.2

Dear Editor,

thank you for giving me the opportunity to revise this manuscript. I have made significant updates and restructuring to the document, according to your suggestion and those of the reviewers. I outline my replies below. Note that the listed commits may not be a complete list of changes to that sections, since later edits were made to the same section. For a complete LaTeX diff, see the attachment.

Changes:

1. *This is not a typical research article since it actually introduces a tool, which is totally fine and I think these kinds of contributions should be accepted in PeerJ CS as well – something I've already done in the past. However, this means that the usual proposed argumentative structure does not work well in this case. I really appreciated the comment by Reviewer 2 who pointed to the ISWC guidelines for resource papers (see https://iswc2019.semanticweb.org/call-for-resources-track-papers/), which can be very helpful for revising the text. See in particular the review criteria at that page, that actually are greatest guidelines also for authors. Thus, I urge to revise the article according to these guidelines so as to make the article more sound for the kind of content it describe.*

The manuscript has been restructured to match a more traditional article, particularly putting part of the article in the correct sections (as suggested by the reviewers). The ISWC guidelines do not provide a clear list of expected sections, but the revision now has the sections Introduction, Background, Approach, Results, Discussion, Outlook, and Conclusion. This commit outlines the changes. Furthermore, the text now more extensively describes the context, related work, etc, as expected by the ISWC guidelines.

2. *An important part that should be addressed, since it is missing in the article, is a related works section where existing works working along the lines of the tool presented in this document are also described and, possibly, compared. This is something that has been indeed requested by all the reviewers. In addition, it would be good to provide also a more general setting of the work by expanding the background section to provide more details.*

The Introduction and Background have been updated and list important prior literature. See this commit.

3. *In the evaluation of the tool, in addition to the study on performances that has been already conducted, and which is interesting, it is needed to discuss the current known uses of the tool. This is a crucial aspect to address since it is the way one understand the community uptake about it. Also, mentions to possible places where the tool can be adopted and which advantages can bring should be discussed as well.*

A section has been added to Results and Table 3. See this commit.

4. *The "Results" and "Discussion" sections must be carefully revised, since they contain a lot of content that should not be present there but rather in other sections.*

I hope the restructuring should sufficiently address this.

Reviewer 1 (Jakob Voß) wrote the following comments.

*The article gives a detailed overview of the Citation.js software with impressive results.*

Thank you.

*The article is a more a report than a research paper and it provides a practical solution to the relevant problem of accessing, and converting bibliographic data for citation. The background section should slightly be extended with references to related works (see comments on validity of the findings).*

Relevant literature is now cited.

*Apart from that there are only minor issues, listed here in their order of appearance :*

● *subsection "output formatting" (line 104) could make more clear that the most of the formatting is done via the existing library citeproc*

This has been clarified in the section Output formatting. See this commit (line 155-156).

● *subsection "CLI use" (line 152) could be reduced as it contains general information common to most command line application*

This has been shortened. See this commit (line 200-206).

● *figure 3 is too large*

This figure was originally smaller, but had to be enlarged to be accepted by the submission system. I have now put back the old image in the document (commit) but will keep the current one in the figure system.

● *there seems to be a "not" missing between line 215 and 216*

This has been clarified in this commit (line 284-285).

● *in the same subsection one might mention that extension of the crosswalk format CSL-JSON might be a solution*

This is now mentioned (commit). I do not know if changing CSL-JSON is viable, since it is not intended as a format for the storage of bibliographic data. At least for some things, the position is that if there are no citation styles requiring it, it probably will not be supported.

- *the subsection on GraphQL (line 235-242) should be removed as it its currently not used in Citation.js and the specific relevance is not made clear*

The subsection has been moved to the new Outlook section, together with some comments to hopefully clarify the relevance (commit, line 320-328).

- *in section line 249-252 one might mention RDF as it is better suitable for merging data from multiple sources that other formats*

Mentioned with this commit (line 302-303).

- *The results are relevant and created in the best form by modern software development, but the research question has been addressed before. The article in should be extended by references to existing solutions and related works. In particular:*
- *mention library catalogues as central stores of bibliographic data (line 29) with library-specific formats such as MARC*

This has been added (commit, line 36-37).

- *mention existing tools: At least Chris Putnam's Bibutils is relevant here: Citation.js seems like a modern successor to Bibutils*

A number of relevant tools have been added (commit, line 82-93).

- *add a subsection about the problem of converting between bibliographic data formats in the background section (with reference to subsection starting at line 198 for details). The topic of schema/metadata crosswalks is relevant both in general information integration and in metadata management in particular. The article does not need to give a full review (see https://en.wikipedia.org/wiki/Schema_crosswalk for a starting point) but it should briefly describe where crosswalks between bibliographic formats already exist (for instance XSLT has played a major role).*

The more expository part of the discussion about crosswalks has been moved to a new subsection in the Background section, together with some added references to earlier work in this area (commit, line 56-81).

Reviewer 2 (Tobias Kuhn):

> *This paper introduces citation.js, which seem to be a very useful and important tool to map between the different formats currently used to represent citation information. The paper is relatively easy to read, but unfortunately not written and structured in a*

*very scientific way and lacks crucial elements. The reader is given many details about the technical details of the tool, but unfortunately virtually nothing is said about its position with respect to the existing scientific literature on the topic of citations, and very little in terms of evaluation results.*

*Specifically, the main sections Background, Results, and Discussion are all very unconventional and lack important scientific content:*

- *The Background section only introduces technologies and libraries that were used to build the tool, but not a broader background in a scientific sense. I expected to read something about other work on analyzing citation networks, on the usage and impact of reference managers, on the properties and uptake of ontologies on scholarly communication, just to mention a few examples from the top of my head.*

The Background has been extended (commit) with information about crosswalks and existing tools.

- *The Results section contains mostly details about the implementation of the tool, which are not results in the scientific sense. These implementation details should rather go into a section called Approach or Implementation or Methods.*

A new section called "Approach" was created with implementation and distribution details (commit).

- *The performance evaluation is the only real result in the Results section, but performance is in my view a secondary problem here. More importantly, I would like to see some results on whether the tool is accurate, useful, and/or actually making an impact.*

The Results section was extended to include subsections describing accuracy (commit) and impact (commit).

- *The Discussion section is also much too technical, whereas I would expect here some more high-level context and interpretation of the presented results.*

The Discussion section was split into a discussion of the approach and setup of the program, and an outlook for possible improvements. Additionally, some technical explanation was moved to the Background (commit).

> *In general, the paper seems to be more of the kind of a Resource Paper as some conferences and journals have them (e.g. here [http://iswc2018.semanticweb.org/call-for-resources-track-papers/](http://iswc2018.semanticweb.org/call-for-resources-track-papers/)), but these papers are typically required to demonstrate and quantify adoption, or at least to provide "evidence and motivation for claiming potential adoption". PeerJ doesn't seem to have an explicit call for such resource papers, but this paper would in any case not meet such typical Resource Paper requirements either.*

*Therefore, in summary, I think that this paper shouldn't be accepted. I have the impression that citation.js is indeed a very useful and important tool, but the omissions of the paper as explained above are too serious to warrant Major Revision as a decision.*

(The ISWC guidelines have been discussed earlier in this letter.)

Reviewer 3 (Sebastian Karcher):

*This is a well written presentation of an important and impressive open-source software contribution and as such I'd recommend it for publication in PeerJ.*

Thank you.

*Most of my review is contained as comments in the uploaded PDF, but some general comments: I don't think the article is well-served by following a structure designed for more "traditional" empirical articles. E.g., it wasn't clear to me why some things were "Results" and some were "Discussion". I would revise this structure, and my reading of PeerJ guidelines allow, if not encourage, authors to do so.*

The structure has been revised. While the Results and Discussion have been kept, the Results are now about findings and the Discussion is a review of the approach.

*The biggest advantage of a journal publication for a software package over simply providing the DOI for the source code itself (which is already the case for citations.js) is that it allows authors to explain their choices and practices in developing the software and to provide a "bigger picture". The present article fully delivers on the former, but I'd like to see some more big picture. For one, a clearer exposition of what exactly the tool does, right at the start of the paper, would help a lot.*

This has been added in the introduction (commit, line 49).

*Secondly, a list of current implementations would be helpful -- I know there's at a minimum Wikicite, but I figure there are others? As it currently stands, the articles requires the reader to be very familiar with the metadata/citation landscape to understand the tool's contribution. I think this can be explained in a way that most readers familiar with academic referencing more broadly could be convinced of its importance.*

A subsection "Impact" has been added (commit).

The reviewer has also provided an annotated manuscript as part of their review:

- "*Citation Style Language (CSL) JSON (Zelle, 2012)*": CSL JSON is not part of the cited CSL specifications. It originates with citeproc-js

The text has been edited to reference CSL variables instead (line 43), citing citeproc-js for CSL-JSON on a later occasion.

- *"Second, existing managers, like Zotero, either require a server or have entirely proprietary backends.": I don't see how this follows. E.g., Zotero can format citations without a server. Maybe I'm misunderstanding, in which case this needs clarification.*

Sorry for the confusion, I seem to have misunderstood too. I meant that Zotero is either used with the website or desktop client (which isn't a server, but does listen to a port for the browser connector) or a translation server (From zotero/translation-server: "The Zotero translation server lets you use Zotero translators without the Zotero client."). The text has been edited to reflect this (line 46-47).

- *"namely with content negotiation.": Given that DOI content negotiation only covers an (albeit very significant) subset of DOIs, e.g. missing the Chinese ISTIC and the EU's publication office, it'd be good to know how Citation.js handles empty results.*

Citation.js currently supports DOIs from CrossRef, DataCite, and mEDRA. Unfortunately, for other DOIs the content negotiation does not result in a proper HTTP 404 error, but with a redirect to the HTML page, which currently results in an unclear error message. I have made an issue to track a fix for that. Additionally, special behavior could be set up for specific registrants, to fetch from e.g. ISTIC and the EU's publication office, assuming the registrant has an API available. I made an issue for that too. In general, empty results *were* handled silently but now throw opt-out errors.

- *"Table 1. Input and output format support. This table only shows general support. For example, the "Wikidata" format is both used for Wikidata identifiers and Wikidata API results.": The paper as a whole remains surprisingly unclear on what exactly citation.js does. I'd turn this table into a figure (see e.g. the graph in the pandoc documentation that illustrates supported conversions as a possible template) and place the figure much earlier in the paper to ensure the reader is clear about the functionality the tool provides from the start.*

A graph and some explanation has been added in the introduction (commit, line 49).

- *"RIS X": This seems wrong? Surely RIS is used as an input, not an output?*

No, RIS is only supported as an output format, not for parsing. Only output was requested, and I haven't gotten around to input yet.

- *"EndNote": Does this refer to Endnote XML or Endnote's proprietary .enl? (I would think reverse engineering the latter runs the risk of running afoul of Endnote's terms of use)*

This refers to the .enw files (line 331).

- *"Machine-Readable Cataloging": As a note, "MARC" is tricky. While the pure machine-readable format exists, many places show instead a human-readable, tabular form and would require reformatting. it may be better to implement Marc XML support instead.*

I intended to support (a version of) the general MARC scheme and whatever format that usually comes in. The text has been changed to say MARC XML instead (line 331).

- *"the Zotero scheme": Does that refer to Zotero RDF, the API JSON, the SQLite database?*

In this case I intended JSON. RDF is on the TODO-list somewhere for the linked-data update (previously planned for 0.5, now for 1.0) (line 331).

- *line 305: "Replace URLs with DOIs in references where available (e.g. here, figshare item below, etc.)"*

This has been done (Zenodo and figshare).

- *line 320-321: "My understanding is that raw.githabusercontent.com may not be stable in the long term. Couldn't you provide a regular gh link instead?"*

The DOI redirects to the raw.githubusercontent.com link but since the DOI isn't visible for this reference type in this citation style I changed it into the DOI URL (commit).

With these changes, I believe to have addressed the important points from the reviewers, and am looking forward to the replies from the reviewers,

with kind regards,

Lars

---

## Round 0.3 · accepted · Accept

Dear author,

Thank you for submitting the revision of your article. The article is now excellent for being published in PeerJ CS. There are only two aspects that should be fixed in your camera ready version of the article:

1. In figure 1, you refer to "citation lists (bibliographies)", while I think it should be "reference lists".

2. You have to restructure the evaluation section in a slightly different way. In particular, there should exist **one** first-level section named "EVALUATION" which should contain an introductory paragraph and two subsections. The first subsection should be called something along the lines of "Design of the evaluation" or "Experimental setting" which introduce what you are going to evaluate – and should contain the current content of your section "Evaluation". The second subsection should be your current "RESULT" section. In practice, the hierarchy of the sections should be as follows:

...

3. APPROACH
3.1 Software Development
...
3.4 Distribution

4. EVALUATION
4.1 Experimental setting
4.2 Results

5. DISCUSSION
...

Have a nice day :-)

S.

---

## Author Rebuttal · Round 0.3

Dear Editor,

I have made the additional updates in reply to the suggestions by the second reviewer. It only deviates from the request in the sense that it only gives a limited set of pointers regarding citation network analysis, which is outside the scope of this work in my opinion. For a complete LaTeX diff, see the attachment.

Changes:

> *The Background section has now a different content than before, partly copied from other places of the previous manuscript, but it is still not a proper background section. It is very technical and very narrow, and thereby fails to give a general and broad overview of this research field. Specifically, this comment from my previous review still holds: "I expected to read something about other work on analyzing citation networks, on the usage and impact of reference managers, on the properties and uptake of ontologies on scholarly communication, just to mention a few examples from the top of my head."*

We thank the reviewer for his comments and we made a few additional general references. However, we believe all the relevant topics have been covered, and this paper is not a review paper. We added more information and citations on reference managers and the distinction between references and citations (commit, line 26–29 and 45–53). Several ontologies and vocabularies were already mentioned, but the BIBO may be of further interest and is mentioned too now. The topic of citation networks is in our opinion outside the scope of this paper. However, we added a reference to relevant literature (commit).

> *As with the Background, also the Approach section lacks a high-level description. The content is all very low-level and technical, which is OK for such an Approach section, but only if preceded by a high-level introduction and explanation. The Approach section start with pointing out what version control system was used, instead of introducing the high-level Citation.js approach.*

A high-level introduction to the approach was added (commit, line 103–119).

> *I like the addition of the accuracy reports in the Results section. This really strengthens the paper!*

Thank you!

> *However, "accuracy" is highly confusing as a term, both intuitively and technically (because of its very specific meaning in statistics). Reporting accuracy levels of 27% makes your tool look performing very poorly when the common interpretation of "accuracy" is applied. I suggest to rename "accuracy" with a more appropriate term, e.g. "coverage". Moreover, the text could make it a bit clearer to what extent these*

*numbers come from a lack of expressiveness of the studied formats or a lack of implemented features on the side of Citation.js (I think it's already in the text to an extent but I found it difficult to understand).*

The term "accuracy" has been replaced with "coverage", and the section has been rewritten into a more logical process of though (commit, line 263–301).

*The designs of the evaluations presented in the Results section should moreover be introduced better (either in a previous section such as Approach, or before the actual results are described in Results). Now, these designs are only described implicitly together with the results. It is good style to separate the two when describing evaluations, i.e. first describing the evaluation's design and only then its results.*

A new section "Evaluation" has been added at the end of the Approach section (commit and commit, line 238–261).

*I like Figure 1! But it's a bit unclear which parts make up the Citation.js tool (all the arrows? only the arrow pointing from "Input" to "Output"?)*

Citation.js is represented by all the arrows, this has been clarified in the figure (commit).

*I like the addition of Impact in the Results section! I think it would help further if Figure 5 was quickly summarized in a sentence (like "Overall, our package was downloaded X times since its creation in October 2016").*

This has been added (commit, line 305–306).

A few minor additional edits have been included. First, since the last revision an archived but authoritative description of RIS has been found in the Internet Archive. Second, the disc tool that was mentioned is now properly cited.

With these changes, I believe to have addressed the remaining points from the second reviewer,

with kind regards,

Lars